# Behavioral engagement patterns and psychosocial outcomes in web-based interpretation bias training for anxiety

Ángel Francisco Vela de la Garza Evia[1]☯*, Jeremy William Eberle[2,3]☯*, Sonia Baee[1], Emma Catherine Wolfe[2], Mehdi Boukhechba[1], Daniel Harold Funk[4], Bethany Ann Teachman[2]*, Laura Elizabeth Barnes[1]

1 Department of Systems and Information Engineering, University of Virginia, Charlottesville, Virginia, United States of America, 2 Department of Psychology, University of Virginia, Charlottesville, Virginia, United States of America, 3 Department of Medical Social Sciences, Northwestern University Feinberg School of Medicine, Chicago, Illinois, United States of America, 4 Sartography, Staunton, Virginia, United States of America

☯ AFV and JWE are joint first authors who contributed equally to this work.
* afv9x@virignia.edu (AFV); jwe4ec@virginia.edu (JWE); bteachman@virginia.edu (BAT)

**Data availability statement:** Data, analysis code, and materials are available at

## Abstract

Digital mental health interventions (DMHIs) have the potential to expand treatment access for anxiety but often have low user engagement. The present study analyzed differences in psychosocial outcomes for different behavioral engagement patterns in a free web-based cognitive bias modification for interpretation (CBM-I) program. CBM-I is designed to shift interpretation biases common in anxiety by providing practice thinking about emotionally ambiguous situations in less threatening ways. Using data from 697 anxious community adults undergoing five weekly sessions of CBM-I in a clinical trial, we extracted program use markers based on task completion rate and time spent on training and assessment tasks. After using an exploratory cluster analysis of these markers to create two engagement groups (whose patterns ended up reflecting generally more vs. less time spent across tasks), we used multilevel models to test for group differences in interpretation bias and anxiety outcomes. Unexpectedly, engagement group did not significantly predict differential change in positive interpretation bias or anxiety. Further, participants who generally spent less time on the program (including both training and assessment tasks) improved in negative interpretation bias (on one of two measures) significantly more during the training phase than those who spent more time (and post hoc tests found were significantly older and slightly less educated). However, participants who generally spent less time had a significant loss in training gains for negative bias (on both measures) by 2-month follow-up. Findings highlight the challenge of interpreting time spent as a marker of engagement and the need to consider cognitive and affective markers of engagement in addition to behavioral markers. Further understanding engagement patterns holds promise for improving DMHIs for anxiety.

https://osf.io/wynxs/ [47]. This manuscript analyzed data collected on or after March 18, 2019, through November 27, 2020 (data collection was deemed over on 12/3/2020; the last datum was collected on 11/13/2020), for participants who enrolled on or before April 6, 2020. The website code for this study is largely covered under the GitHub tags from Version 2.1 through Version 2.12, which include code fixes and changes; for Version 2.12, see https://github.com/TeachmanLab/MT-Java/tree/v2.12.

**Funding:** A grant awarded to BAT and LEB (R01MH113752) and a grant supporting JWE (T32MH115882) from the National Institute of Mental Health (https://www.nimh.nih.gov/) funded this work. The funder had no role in study design, data collection and analysis, decision to publish, or preparation of the manuscript.

**Competing interests:** The authors have declared that no competing interests exist.

## Author summary

Digital mental health interventions can increase access to care for the many people with anxiety facing treatment barriers. For example, online programs that train people to interpret ambiguous situations in less threatening ways can reduce anxiety symptoms, without requiring a psychotherapist. However, many who start such programs do not engage with them much (based on tasks completed and time invested), which may limit program effectiveness. We assessed the impact of anxious adults' engagement patterns in a five-session web-based interpretation bias training program on their improvements in anxious thinking and anxiety symptoms. We used behavioral engagement data (i.e., number of tasks users completed, time users spent on tasks) to create two user groups, reflecting those who generally spent more versus less time on both training and assessment tasks. Unexpectedly, for most treatment outcomes, users who spent more time did not improve significantly more than users who spent less time, and for one outcome users who spent less time improved more. Findings highlight the challenge of interpreting time spent as a measure of engagement and the need to also measure how users think and feel about the program as they complete tasks to more fully understand how engagement patterns impact treatment outcomes.

## Background and significance

The prevalence of clinically significant anxiety has grown worldwide, leading to increased demand for mental health services [1]. However, multiple barriers, such as limited availability and high cost of in-person therapy, prevent individuals from receiving adequate services [2]. Digital mental health interventions (DMHIs) have the potential to reduce these barriers and increase access to care [3–5]. For example, web-based cognitive bias modification programs that target interpretation biases (CBM-I) associated with anxiety [6,7] can shift the tendency of anxious people to assign threatening meanings to ambiguous situations by providing practice resolving ambiguous scenarios positively, without requiring a therapist [8].

Although DMHIs for anxiety can yield small-to-moderate reductions in symptoms, high rates of attrition and low user engagement may limit the efficacy of these interventions and challenge their widespread adoption [9–12]. These challenges also apply to CBM-I, which can improve anxiety symptoms but has yielded somewhat mixed findings [13] and high attrition rates in clinical trials [14,15]. Understanding the impact of users' engagement patterns on their CBM-I outcomes may shed light on these mixed findings, clarify dose-response relationships, identify different patterns of effective engagement and mechanisms of change, and guide efforts to tailor CBM-I to users' needs [16–20].

Engagement in DMHIs is a multifaceted construct with at least two parts: (a) extent of program use (i.e., *behavioral engagement*), and (b) the user's subjective experience throughout the program [21], which involves both cognitive and affective aspects [22]. Most research has analyzed extent of use, which has been assessed with behavioral metrics (e.g., time spent online, completion rate, frequency of use) from questionnaires, ecological momentary assessments, sensors, or system logs [18,20]. Greater use on a given metric is typically expected to yield better outcomes [23], but mixed findings and methodological differences across studies [24] highlight the need for further research analyzing engagement as a more complex construct. In particular, examining correlates of use, such as symptom severity and demographic characteristics, may help reveal the meaning of users' behavioral engagement (e.g., lack of time spent on a program likely has a different meaning when a user is higher vs. lower in symptom severity), providing a fuller picture of engagement patterns.

The dominant approach to studying the effect of engagement on outcomes in DMHIs has been to analyze associations between individual engagement metrics (or composites of metrics) and outcomes [25–30]. Another approach, which we take in the present paper, has been to group users by multiple engagement metrics (e.g., using data-driven cluster analysis) and analyze whether these groups show differential symptom reduction [16,17,19,30–32]. The former approach is *variable centered*, focused on average relations between certain variables and outcomes in a single population. By contrast, the latter approach is *person centered*, focused on specific configurations of values across a set of variables, possibly but not necessarily representing multiple subpopulations, and on the effects of these configurations on outcomes [33,34].

Using cluster analysis to create engagement subgroups is promising when there are multiple possible engagement features because cluster analysis's use of multiple (vs. single or composite) metrics and its exploratory nature can suggest clusters that differ based on potentially more complex engagement patterns in the data than can be represented by single metrics or predicted a priori [16,17,19,35]. For example, exploratory classification methods have identified Low Engagers, Late Engagers, High Engagers With Rapid Disengagement, High Engagers With Moderate Decrease (in Engagement), and Highest Engagers, with certain group differences in symptom outcomes [16]. This person-centered approach may be especially useful for studying real-world engagement in implementation settings, where users' varied expectations, goals, needs, and resources may yield more complex engagement patterns than those in efficacy trials [19].

## Objective

The present study analyzed whether behavioral engagement subgroups, created using an exploratory cluster analysis of program use markers (i.e., task completion rate, time spent on training and assessment tasks), show differential improvement in anxiety and interpretation bias in a trial of web-based CBM-I administered to anxious adults on our team's public research website *MindTrails*. Although we hypothesized that the group(s) with metrics suggesting higher engagement would have significantly better outcomes in general, we did not have hypotheses about the specific (and potentially complex) engagement patterns that the cluster analysis might suggest. After the cluster analysis suggested two groups differing on time spent across tasks, we expected the group that generally spent more time doing the program (including both training and assessment tasks) would improve more (based on the purported positive relation between extent of use and outcomes [23], and on the positive relation in a prior MindTrails study between mean time spent per CBM-I scenario and anxiety improvement) [28]. However, we also recognized plausible alternatives (e.g., taking longer can indicate distraction) [24] and ran post hoc comparisons of the groups on various baseline variables (e.g., demographics) to more fully characterize them. (Given that another post hoc test indicated that the existence of two groups was not significantly more plausible than the existence of one group, we consider the two groups descriptive subgroups of a single population rather than two truly distinct subpopulations [36].)

## Materials and methods

### Participants and procedure

We analyzed data from 697 community adults with at least moderate anxiety symptoms ($\geq 5$ on an adapted version of the Anxiety Subscale of the Depression, Anxiety, Stress Scales-Short Form [37]; DASS-21 AS; i.e., $\geq 10$ on DASS-42 AS) who had been randomly assigned to the

CBM-I condition of the Calm Thinking study [15], a multistage randomized trial (*Clinical-Trials.gov* ID: NCT03498651) run on the MindTrails website (https://mindtrails.virginia.edu). We focused on participants who started the first of five weekly training sessions. Out of the 984 who started the first session, we excluded a subgroup who were later randomly assigned to receive supplemental telecoaching as part of the parent study (*n* = 282; see Section A.1 in S1 File), those with repeated eligibility screenings (*n* = 3), and those identified as outliers in most of the engagement markers (*n* = 2). For a flow diagram, see Fig 1. (Starting training was defined as viewing instructions on the page before the first CBM-I scenario. Given that 70 of the 697 participants stopped at the instructions page and did not complete any scenarios, as a

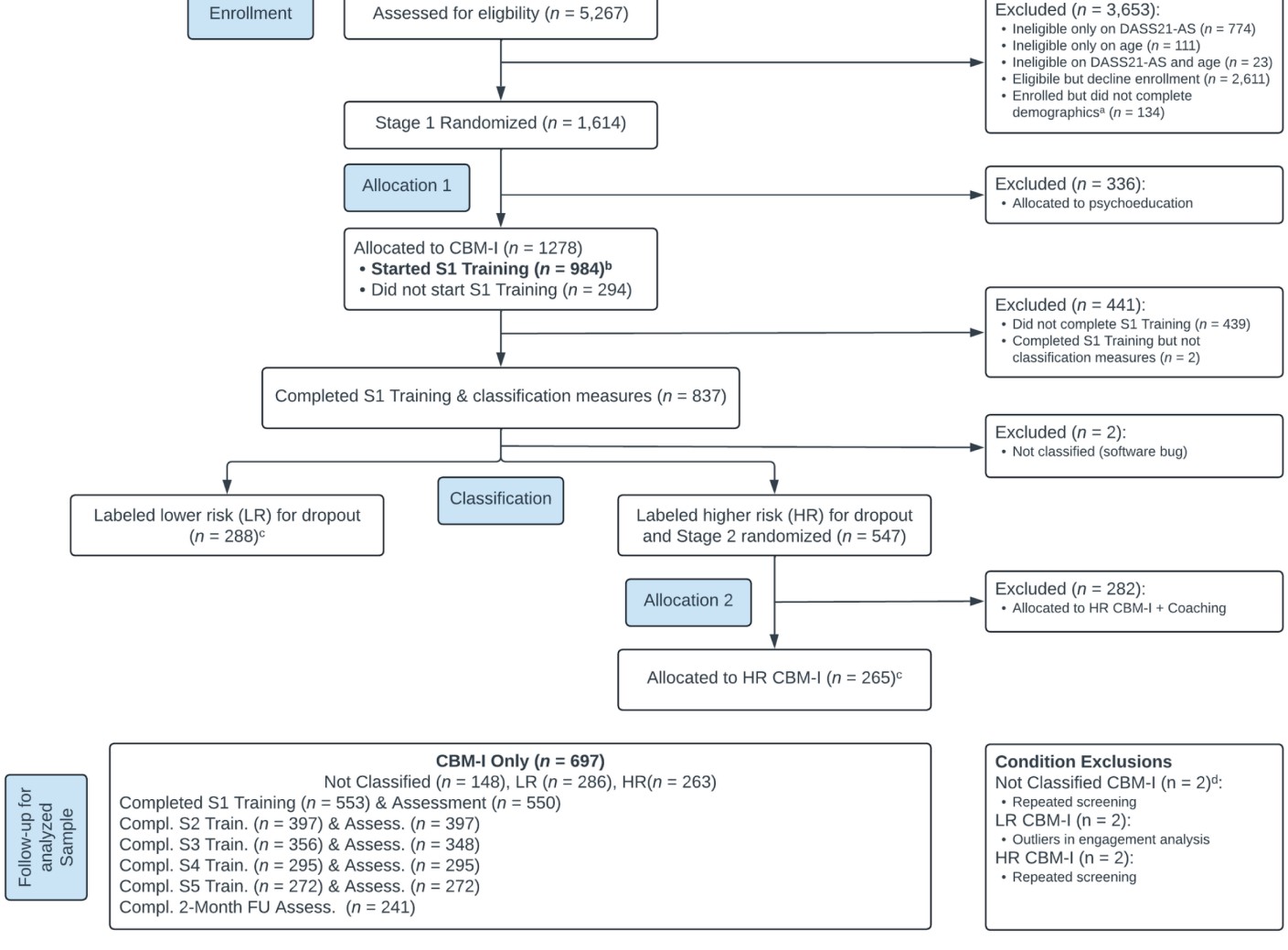

**Fig 1. Consort diagram.** *Classification Note.* The parent Calm Thinking study attempted to use an algorithm to classify CBM-I participants as lower- vs. higher- risk of dropout so that higher-risk participants could be randomized to either receive supplemental telecoaching or continue with only CBM-I. However, the accuracy of the algorithm was not better than chance, suggesting that the risk classification does not reflect meaningful differences in risk. Thus, the "Lower risk (LR) for dropout" and "Higher risk (HR) for dropout" participants have comparable dropout risk, and the present study's exclusion of "HR CBM-I + Coaching" participants is an exclusion of participants assigned to an additional intervention, not an exclusion of participants at meaningfully higher risk. For details, see Calm Thinking main outcomes paper. *Note.* Adapted from the Calm Thinking main outcomes paper CONSORT diagram. Analysis exclusions are included in flow but not analyzed. S1-5 = Session 1-5; FU = Follow-Up. [a]needed for stratification. [b]analyzed sample before exclusions. [c]condition classification count before exclusions. [d]1 did not start S1 training; 1 started S1 training but did not complete classif. measures.

sensitivity analysis we reran analyses excluding these users [29]. Results were nearly identical [see Tables B.5-B.7 and Fig C.5 in S1 File], so we focus on the original sample.)

Enrollment opened on March 18, 2019, and closed on April 6, 2020; we analyzed data collected through November 27, 2020. After providing informed consent (by clicking a button indicating that they have read the consent form and agree to participate), participants were asked to complete assessments at pretreatment, after each training session, and at 2-month follow-up. At least 5 days had to pass between finishing one session's assessment and starting the next session's training; a mean of 9.93-11.53 days elapsed between consecutive training sessions (see Table B.11 in S1 File). At least 60 days had to pass between completing Session 5's assessment and starting the follow-up assessment. Participants received $5 in gift cards after completing each of the pretreatment, Session 3, and Session 5 assessments and $10 after completing the follow-up assessment. The University of Virginia Institutional Review Board for the Social and Behavioral Sciences approved all procedures. For more details about this trial, see the main outcomes paper [15].

## Cognitive bias modification for interpretation (CBM-I)

Each CBM-I session, designed to take about 15 min, consisted of a unique set of 40 ambiguous scenarios about everyday situations (e.g., social settings, physical sensations, health concerns) that could prompt anxiety. After seeing a scenario's title (e.g., "Spotting a Neighbor") and image, participants read the description, which ended with a word fragment that resolved the ambiguity as positive 90% of the time and negative otherwise (e.g., "As you are walking down a crowded street, you see your neighbor on the other side. You call out, but they do not answer you. Standing there in the street, you think that this must be because they were distr_cted" [positive]). Participants selected one or more letters to complete the word fragment, which increased in difficulty across the five sessions. Participants then answered a comprehension question that reinforced the scenario's meaning and whose format varied. (For details, see Section A.2 in S1 File.)

## Outcome measures and covariate

### Anxiety symptoms.

*Overall Anxiety Severity and Impairment Scale (OASIS).*

The Overall Anxiety Severity and Impairment Scale (OASIS), our primary measure of anxiety, is a five-item self-report of the frequency and severity of anxiety and avoidance and the degree of related impairment in the past week (adapted from the original version [38]; see Section A.3 in S1 File for measure details and adaptations). Participants rate each item on a 5-point scale from 0 (*lowest frequency/severity*) to 4 (*highest frequency/severity*). The OASIS was assessed at baseline (pretreatment), after Sessions 1-5, and at follow-up. Internal consistency for the analyzed sample using complete item-level data at baseline was good, Cronbach's $\alpha = .83$.

*Depression, Anxiety, Stress Scales-Short Form: Anxiety Subscale (DASS-21 AS).*

The DASS-21 AS, our secondary measure of anxiety, is a seven-item self-report of the frequency of anxiety symptoms (especially physical symptoms; adapted from the original version) [37]. Participants rate the extent each item applied to them in the past week on a 4-point scale from 0 (*not at all*) to 3 (*most of the time*). The DASS-21 AS was assessed at baseline (screening), after Sessions 3 and 5, and at follow-up. Internal consistency at baseline was acceptable, $\alpha = .73$.

**Interpretation bias.**

*Recognition Ratings (RR).*

Recognition Ratings (RR, developed by our team to resemble those in the original version [39]) is our primary measure of interpretation bias. Participants read nine ambiguous scenarios and then for each scenario rate how similar each of four possible interpretations is to the scenario on a 4-point scale from 1 (*very different*) to 4 (*very similar*). For details, see Section A.4 in S1 File. Negative and positive bias scores were calculated by averaging ratings of negative and positive threat-relevant interpretations, respectively (per a prior paper) [14]. RR was assessed at baseline (pretreatment), after Sessions 3 and 5, and at follow-up. Internal consistency at baseline was acceptable for both scores, $\alpha$s = .74 and .73, respectively.

*Brief Body Sensations Interpretations Questionnaire (BBSIQ).*

The Brief Body Sensations Interpretations Questionnaire (BBSIQ, modified from the original version [40]) is our secondary measure of interpretation bias. Participants read 14 ambiguous situations and rate the likelihood of three explanations using a 5-point scale from 0 (*not at all likely*) to 4 (*extremely likely*). For details, see Section A.5 in S1 File. A negative bias score was calculated by averaging the ratings for all negative, threat-relevant explanations [14]. The BBSIQ was assessed at baseline (pretreatment), after Sessions 3 and 5, and at follow-up. Internal consistency at baseline was excellent, $\alpha$ = .90.

**Training confidence.**

*Item from Readiness Ruler.*

In the multilevel model below, we included one item that assessed confidence in online anxiety programs from the Readiness Ruler [41] at baseline as a covariate given that higher confidence was associated with lower dropout and greater improvement in a prior MindTrails study [42]. Participants rated "How confident are you that an online training program will reduce your anxiety?" on a scale from 0 (*not at all*) to 4 (*very*).

## Engagement markers

We defined program use markers for a cluster analysis (see below) based on task completion rate and time spent on training and assessment tasks. Some prior studies have also treated assessment tasks as relevant to engagement, using rates of completing assessment tasks as engagement markers [18] or including time spent on assessment tasks in predictors of dropout [43] or treatment outcomes [44,45]. We viewed assessment tasks as relevant to engagement in the program for a few reasons. First, interventions can include assessments of their targets and outcomes solely to improve efficacy by providing progress feedback [46] (which users were told they would receive for their OASIS scores at the end of the parent trial) or by eliciting desirable reactive effects from the assessment task itself, even without feedback [47]. Second, the training and assessment tasks, some of which had similar formats (e.g., the CBM-I training task and RR and BBSIQ assessment tasks all present ambiguous scenarios), were slightly intermixed and presented seamlessly at training sessions; thus, users may not have viewed all training and assessment tasks as distinct. Third, interventions can include assessments to adapt treatment based on certain tailoring variables [48], as the parent trial used self-reported responses (i.e., demographics, state anxiety) and passive features (e.g., time spent) from assessment tasks to classify users at risk of dropout [15]. Given that we did not

view training and assessment tasks as fully independent, we allowed all tasks to inform the clusters' potentially complex engagement patterns. (For an initial cluster analysis based on task completion rate but fewer time-related markers, see Section A.6.1 in S1 File.)

**Task completion rate.** Each training session and its associated assessment measures were designed to be completed in one sitting; on average, participants logged in once per session. All participants were asked to complete 5 training sessions and 63 assessment measures, for a total of 68 tasks. Each participant's task completion rate was calculated as the proportion of tasks that were completed, similar to other papers [17].

**Time on training components.** We computed time spent (typically by adding reaction times from the start of an exercise to whatever point in the exercise the participant reached) on the following components of each training session, similar to other papers [43]. For participants with repeated entries for an element within a given component (e.g., a page within an exercise), we removed any duplicated entries and then computed the average reaction time for that element.

*Time on imagery practice exercise.*

This exercise was given at the start of the first training session to prepare participants to vividly imagine CBM-I scenarios. Participants were asked to use all senses to imagine what it feels like to hold a lemon in their hand.

*Time on anxiety imagery prime exercise.*

Next, in the first training session, participants were asked to imagine themselves in an anxiety-provoking situation that they were likely to experience to activate their anxious thinking about a personally relevant situation.

*Mean time per CBM-I scenario across sessions.*

The final training component in the first session (and only training component in later sessions) was 40 CBM-I scenarios. First, time spent on an individual scenario was calculated by summing reaction times across its three parts (title and image, description, comprehension question). For the description and question, we included the time taken to submit a correct response. Then, we averaged the time spent on completed scenarios for a given session. Finally, we used those values to compute the mean time spent on an individual scenario across sessions for a given participant [28].

**Time on assessment measures.** We analyzed time spent on measures assessed before the start of Session 1 training so that all participants would have at least one data point. These measures are in Table 1 and included both one-time measures (e.g., demographics, mental health history) and repeated measures (e.g., interpretation biases, anxiety and comorbid symptoms, other cognitive mechanisms, wellness) [43]. For measure details, see Section A.7 in S1 File.

For repeated measures, the average time spent across completed administrations of that measure was computed. To avoid collinearity, we computed Pearson product-moment correlations for time spent values across measures and excluded time spent values with correlations greater than .70. Time spent on BBSIQ was excluded given its high correlation with time spent on RR, Mechanisms, and Wellness measures. For correlations among all engagement features, see Table B.13 in S1 File.

**Table 1. Assessment measures used for 'Time Spent' engagement markers.**

| Task (From Task Log) | Description | Session | Number of Items |
|---|---|---|---|
| Anxiety Symptoms (DASS-21 AS) | Assesses anxiety symptoms (DASS-21 AS) | SC; after S3 and S5; FU | 7 |
| Credibility | Assesses importance of reducing anxiety (Importance Ruler) and confidence in intervention [41] | PTX | 2 |
| Demographics | Assesses age, gender, race, ethnicity, education, employment status, marital status, income, country | PTX | 10 |
| Mental Health History | Assesses mental health disorders and treatments | PTX | 4 |
| Anxiety Identity | Assesses centrality of anxiety to identity (Anxiety and Identity Circles) | PTX; after S3 and S5; FU | 1 |
| Anxiety Symptoms (OASIS) | Assesses anxiety symptoms (OASIS) | PTX; after S1-5; FU | 5 |
| Anxiety Triggers | Assesses situations that prompt anxiety | PTX | 6 |
| Interpretation Bias (RR) | Assesses positive and negative interpretations of ambiguous situations | PTX; after S3 and S5; FU | 36 |
| Interpretation Bias (BBSIQ) | Assesses positive and negative interpretations of ambiguous situations | PTX; after S3 and S5; FU | 42 |
| Depression and Alc. Use (Comorbid) | Assesses depression symptoms (PHQ-2) and alcohol use (AUDIT-C) | PTX; after S3 and S5; FU | 5 |
| Wellness | Assesses self-efficacy (NGSES), growth mindset (PBS), optimism (LOT-R), and life satisfaction [49] | PTX; after S3 and S5; FU | 9 |
| Mechanisms | Assesses cognitive flexibility (CFI), experiential avoidance (CompACT), cognitive reappraisal (ERQ), and intolerance of uncertainty (IUS-12) | PTX; after S3 and S5; FU | 6 |
| Technology Use | Assesses frequency of use for different device types | PTX | 4 |
| Affect | Assesses state anxiety (SUDS) | Before S1, S3, and S5 | 1 |

*Note.* Tasks are listed top to bottom in the order they were first administered in the study (i.e., DASS-21 AS at SC; then tasks at PTX, in the order they were administered; then Affect before S1); additional time points are listed for repeated measures. The number of items for each measure at a given session is shown. DASS-21 AS = Depression, Anxiety, Stress Scales-Short Form: Anxiety Subscale; SC = Screening; S# = Session Number; FU = follow-up; PTX = pretreatment; OASIS = Overall Anxiety Severity and Impairment Scale; RR = Recognition Ratings; BBSIQ = Brief Body Sensations Interpretations Questionnaire.

## Statistical analysis

Data were obtained from the Public Component (https://osf.io/s8v3h/) of the MindTrails Calm Thinking Study's OSF project, outputted from the study's central cleaning scripts (ver. 1.0.0) [50]. All analyses except multiple imputation (see below) were done in R (ver. 4.1.1) [51]. All analyses used an alpha level of .05. Data and analysis code are available at the project's OSF page (https://osf.io/wynxs/) [52].

**Cluster analysis.**

*Outliers, transformations, and missing data handling for engagement markers.*

Before conducting the cluster analysis, we excluded outlying values of time-related engagement markers; log-transformed time-related markers (except for time spent on CBM-I scenarios); and standardized all markers, including completion rate (see Section A.8 in S1 File). For details on missing data handling, see Section A.9 in S1 File.

*K-means clustering.*

To analyze variability in (potentially complex) engagement patterns, we explored different clustering algorithms (unsupervised learning methods that can suggest subgroups in multidimensional data) to group participants based on their engagement markers. After considering *K*-means clustering, partitioning around medoids, and agglomerative hierarchical clustering while specifying two to four clusters (for an overview of these algorithms, see Section A.6.2 in S1 File), we chose *K*-means because of its creation of clusters with similar sample sizes, its superior validation indices, and its popularity (see Section A.10 in S1 File). We used the NbClust package (ver. 3.0) [53] to compute the optimal number of clusters and the kmeans function of the stats package (ver. 4.3.1) [54] to create the clusters. The optimal number of clusters was two.

However, a homogeneity test (Steinley & Brusco's lower bound ratio, LBR, test) [55] run post hoc did not support the presence of more than one cluster. Specifically, the ratio of the within-cluster sum of squares for the 2-means solution to the sum-of-squares total ($SSE_2/SST$ = .71) was not less than the lower bound of the ratio that is obtainable for splitting a multivariate normal distribution in half (i.e., .36). (Although a Duda-Hart test [56] supported two clusters, we focus on the more conservative LBR test; see Section A.11 in S1 File.) Thus, our clusters are best interpreted as two descriptive subgroups of a single population rather than as two truly distinct subpopulations [36,57]. Notably, a distinction between descriptive ("constructivist") and inferential ("realist") goals of a given cluster analysis is not straightforward (nor is the definition of a "real" cluster), and clusters can be used for pragmatic reasons (e.g., exploratory analysis of data patterns and the effects of the patterns on other variables) even if the clusters do not stem from "real" subpopulations (see this review [57]). We view our clusters as useful partitions of a high-dimensional space of engagement features that allow us to analyze the effects of variability in that complex space on treatment outcomes.

To verify that the clusters differed on engagement markers (following [17]), which were not normally distributed [17], we conducted Wilcoxon rank-sum nonparametric tests using the wilcox.test function in the stats package (ver. 4.3.1) [51]. To further characterize the clusters after analyzing their differences in outcome trajectories (see below), we also ran post hoc tests of their baseline differences on selected demographic variables, anxiety symptom and interpretation bias outcomes, comorbid symptoms (depression, Patient Health Questionnaire-2 [58]; alcohol use, Alcohol Use Disorders Identification Test-Concise) [59], and training confidence. We computed the effect size ($r$) for each rank-sum test [60] using the wilcox_effsize function of the rstatix package (ver. 0.7.2) [61]. For tests of group differences on categorical demographic variables, we conducted chi-squared tests using the chisq.test function of the stats package and computed effect sizes as Cramér's $V$ [62]. We interpreted $r$ and Cramér's $V$ per Cohen's (1988) guidelines [62].

**Multilevel modeling**

*Missing data handling for outcome measures and imputation model.*

For details on missing data handling for outcomes (see Table B.2 in S1 File for descriptives for each outcome over time), including fully Bayesian multiple imputation based on our analysis model [63], see Section A.12 in S1 File.

*Analysis model.*

To analyze the effect of engagement group on outcome trajectories, we fit a separate multilevel model [64] in each of the imputed datasets for each outcome using the nlme package (ver. 3.1-152) [65] and pooled results across datasets using the mitml package (ver. 0.4-3) [66], which follows Rubin's rules [67]. Time (assessment point) was represented as two piecewise linear trajectories: $time_{TR}$ during the training phase (baseline to Session 5; coded as 0 for baseline, as 1-5 for Sessions 1 through 5, and as 5 for follow-up) and $time_{FU}$ during the follow-up phase (Session 5 to follow-up; coded as 0 for baseline through Session 5 and as 1 for follow-up). (For specific time codings for each outcome, see Section A.13 in S1 File.) Each model included the fixed effects of engagement group (dummy coded as 0 for less time spent and 1 for more time spent), $time_{TR}$, $time_{FU}$, Engagement Group × $Time_{TR}$, Engagement Group × $Time_{FU}$, and the training confidence covariate (grand mean centered); a random intercept; and random slopes for $time_{TR}$ and $time_{FU}$. We first interpreted the Engagement Group × Time interactions. If one was significant, we analyzed the simple effects of time with

a separate model in each group that included the fixed effects of time$_{TR}$, time$_{FU}$, and training confidence; a random intercept; and random slopes for time$_{TR}$ and time$_{FU}$. For details, see Section A.16 in S1 File. Parameter estimates are reported as unstandardized $b$ with 95% confidence intervals (CIs) from the confint function.

*Effect size.*

For all outcomes, we computed the standardized mean difference between engagement groups at Session 5 and at follow-up as growth modeling analysis (GMA) $d$ [68] using the pooled standard deviation at baseline. Given that the groups were not randomized, we computed GMA $d$ unadjusted for baseline mean differences in addition to GMA $d$ adjusted for such differences. GMA $d$ unadjusted for baseline differences reflects group differences in both initial status at baseline and growth from baseline to a given time point [69,70]. GMA $d$ adjusted for baseline differences (which is equivalent to Cohen's $d$) [70] reflects group differences in only growth. For within-group effect sizes, we computed each group's standardized mean difference from baseline to Session 5 and to follow-up using the group's standard deviation at baseline (A. Feingold, personal communication, March 3-4, 2019).

## Results

### Demographic characteristics

Most participants ($M$ age = 35 years) identified as female (80.9%), White/European origin (70.4%), Not Hispanic or Latino (81.8%), and from the United States (91.7%). For full demographic characteristics and for baseline levels of anxiety symptoms and interpretation biases for the analyzed sample and each engagement group, see Tables B.1 and B.2 in S1 File, respectively.

### Cluster analysis

The two clusters significantly differed on all time-related markers and not on task completion rate (see Table 2). Visual inspection of each group's distribution for time spent on training (see Fig C.1 in S1 File) and assessment (see Figs C.2 and C.3 in S1 File) tasks and for task completion rate (see Fig C.4 in S1 File) revealed that one group spent less time on training tasks (with small-medium effect sizes: $r$s = .24-.49; Table 2) and assessment tasks (with large effect sizes: $r$s = .56-.73) than the other group (whereas completion rates were balanced). Thus, we labeled the groups as "Less Time Spent" ($n$ = 386) and "More Time Spent" ($n$ = 311), respectively.

Notably, although the clusters differ on average on every time-related feature, this does not mean that all participants in the Less (More) Time Spent group have lower (higher) values than all participants in the other group on every feature; as shown in Figs C.1-C.3 in S1 File, the groups' distributions on a given feature overlap. As such, the two clusters are not the result of partitioning on every feature individually (e.g., akin to applying a median split to every feature, which would likely face the issue of participants' falling below the median on some features but above on others). Rather, the clusters are the result of partitioning on all 17 features at once in a 17-dimensional space. Note that this person-centered approach, which allows us to contrast two complex, descriptive patterns of engagement on treatment outcomes, also differs from a variable-centered approach that aggregates across the time-related features. For example, analyzing whether the mean of all time-related features moderates outcomes would address a different question by reducing the multidimensional space to one number (and lose our focus on potentially complex patterns of engagement).

**Table 2. Tests of group differences in engagement markers.**

| Engagement Marker | Less Time Spent (n = 386) Median (IQR) | More Time Spent (n = 311) Median (IQR) | Wilcoxon rank-sum test | p | r |
|---|---|---|---|---|---|
| **Task completion rate** | 0.57 (0.75) | 0.49 (0.69) | 58,520.0 | .560 | .02 |
| **Time on training components** | | | | | |
| Time on imagery practice exercise (min) | 0.88 (0.61) | 1.44 (0.91) | 31,077.0 | <.001 | .42 |
| Time on anxiety imagery prime exercise (min) | 1.66 (0.82) | 2.54 (1.40) | 25,833.0 | <.001 | .49 |
| Mean time per CBM-I scenario across sessions (sec) | 11.18 (6.19) | 13.88 (5.56) | 43,202.5 | <.001 | .24 |
| **Mean time on assessment tasks (min)** | | | | | |
| Anxiety (DASS-21 AS)[a] | 0.53 (0.22) | 0.83 (0.42) | 17,764.5 | <.001 | .61 |
| Credibility | 0.26 (0.31) | 0.85 (0.65) | 19,807.0 | <.001 | .58 |
| Demographics | 1.05 (0.38) | 1.72 (0.76) | 15,187.0 | <.001 | .64 |
| Mental Health History | 1.07 (0.50) | 1.97 (1.07) | 13,317.5 | <.001 | .67 |
| Anxiety Identity[a] | 0.19 (0.10) | 0.32 (0.16) | 18,250.0 | <.001 | .60 |
| Anxiety (OASIS)[a] | 0.45 (0.15) | 0.73 (0.31) | 11,735.5 | <.001 | .69 |
| Anxiety Triggers | 0.66 (0.33) | 1.23 (0.67) | 14,713.0 | <.001 | .65 |
| Interpretation Bias (RR)[a] | 2.58 (0.91) | 4.16 (1.53) | 11,246.5 | <.001 | .70 |
| Depression and Alc. Use (Comorbid)[a] | 0.43 (0.21) | 0.71 (0.30) | 16,564.5 | <.001 | .62 |
| Wellness[a] | 0.70 (0.28) | 1.24 (0.50) | 9,420.5 | <.001 | .73 |
| Mechanisms[a] | 0.55 (0.24) | 0.89 (0.45) | 14,127.5 | <.001 | .66 |
| Technology Use | 0.25 (0.11) | 0.40 (0.19) | 21,272.0 | <.001 | .56 |
| Affect[a] | 0.13 (0.06) | 0.21 (0.12) | 20,579.0 | <.001 | .57 |

*Note.* Time spent on the Brief Body Sensations Interpretations Questionnaire was excluded given its high correlation with time spent on RR, Mechanisms, and Wellness measures. IQR = interquartile range; DASS-21 AS = Depression, Anxiety, Stress Scales-Short Form: Anxiety Subscale; OASIS = Overall Anxiety Severity and Impairment Scale; RR = Recognition Ratings. For measure details, see Table 1. Per Cohen (1988), $rs \geq .1, .3,$ or .5 are small, medium, or large, respectively [62].
[a]Repeated measure.

Post hoc tests revealed that the groups also significantly differed on age, race, education, anxiety symptoms (on DASS-21 AS but not OASIS), and negative interpretation bias (on both RR and BBSIQ; see Table B.4 in S1 File). The group that spent less time on the program (including both training and assessment tasks) was younger, identified with multiple races (vs. one) at a higher rate, and had higher levels of education (see Table B.1 in S1 File) and higher baseline anxiety and negative bias (see Table B.2 in S1 File). These effect sizes were medium for age ($r = .30$; see Table B.4 in S1 File), small for race (Cramér's $V = .16$), very small for education and anxiety ($rs = .08-.09$), and very small or small for negative bias ($rs = .08-.12$). The groups did not significantly differ on gender, ethnicity, positive bias, depression symptoms, alcohol use, or training confidence (see Table B.4 in S1 File).

## Multilevel modeling

**Anxiety symptoms.** Contrary to our hypothesis, the slopes of the engagement groups' trajectories for anxiety symptoms (assessed by OASIS and DASS-21 AS) did not significantly differ during the training or follow-up phases (Table 3).

**Positive interpretation bias.** Also unexpectedly, the slopes of the groups' trajectories for positive interpretation bias (assessed by RR) did not significantly differ during the training or follow-up phases (Table 3).

**Negative interpretation bias.** For negative interpretation bias assessed by RR, the slopes of the engagement groups' trajectories significantly differed during the training phase ($b = 0.03, p = .040$; Table 3). As shown in Fig 2, both groups had significant reductions from baseline to Session 5. However, unexpectedly, the group that generally spent less time on the

**Table 3. Piecewise linear multilevel modeling fixed effects.**

| Outcome | Fixed Effect | b (SE) | t | df | p | 95% CI | d (Adj.)[a] | d (Unadj.)[b] |
|---|---|---|---|---|---|---|---|---|
| OASIS | Intercept | 2.25 (0.04) | 62.49 | 3,492.21 | <.001*** | [2.18, 2.32] | | |
| | time$_{TR}$ | −0.15 (0.01) | −14.07 | 320.50 | <.001*** | [−0.17, −0.13] | | |
| | time$_{FU}$ | 0.12 (0.06) | 2.01 | 250.32 | .045* | [0.00, 0.25] | | |
| | More Time Spent | 0.07 (0.05) | 1.38 | 641.69 | .169 | [−0.03, 0.18] | | |
| | Training confidence | 0.01 (0.03) | 0.27 | 535.14 | .789 | [−0.05, 0.07] | | |
| | More Time Spent × time$_{TR}$ | −0.01 (0.02) | −0.68 | 306.56 | .494 | [−0.04, 0.02] | −0.08 | 0.03 |
| | More Time Spent × time$_{FU}$ | 0.00 (0.11) | 0.03 | 180.43 | .975 | [−0.21, 0.21] | −0.08 | 0.03 |
| DASS-21 AS | Intercept | 1.63 (0.03) | 59.82 | 2,022.03 | <.001*** | [1.58, 1.69] | | |
| | time$_{TR}$ | −0.15 (0.01) | −16.29 | 232.80 | <.001*** | [−0.16, −0.13] | | |
| | time$_{FU}$ | −0.01 (0.05) | −0.30 | 157.29 | .764 | [−0.11, 0.08] | | |
| | More Time Spent | −0.09 (0.04) | −2.15 | 673.94 | .032* | [−0.17, −0.01] | | |
| | Training confidence | 0.02 (0.02) | 0.76 | 503.96 | .449 | [−0.03, 0.07] | | |
| | More Time Spent × time$_{TR}$ | −0.02 (0.01) | −1.19 | 249.28 | .234 | [−0.04, 0.01] | −0.15 | −0.31 |
| | More Time Spent × time$_{FU}$ | 0.06 (0.06) | 1.00 | 190.09 | .320 | [−0.06, 0.19] | −0.03 | −0.19 |
| BBSIQ | Intercept | 1.57 (0.04) | 37.09 | 2,033.33 | <.001*** | [1.49, 1.65] | | |
| | time$_{TR}$ | −0.17 (0.01) | −13.46 | 353.23 | <.001*** | [−0.19, −0.14] | | |
| | time$_{FU}$ | 0.16 (0.05) | 3.40 | 197.54 | .001** | [0.07, 0.26] | | |
| | More Time Spent | −0.13 (0.06) | −2.03 | 677.37 | .043* | [−0.25, −0.00] | | |
| | Training confidence | 0.04 (0.03) | 1.11 | 354.69 | .270 | [−0.03, 0.10] | | |
| | More Time Spent × time$_{TR}$ | 0.01 (0.02) | 0.74 | 336.31 | .463 | [−0.02, 0.05] | 0.08 | −0.07 |
| | More Time Spent × time$_{FU}$ | −0.15 (0.07) | −2.14 | 213.62 | .033* | [−0.29, −0.01] | −0.09 | −0.25 |
| RR Negative Bias | Intercept | 2.92 (0.03) | 110.20 | 1,969.37 | <.001*** | [2.87, 2.97] | | |
| | time$_{TR}$ | −0.10 (0.01) | −10.3 | 275.16 | <.001*** | [−0.12, −0.08] | | |
| | time$_{FU}$ | 0.13 (0.04) | 3.09 | 201.30 | .002** | [0.05, 0.21] | | |
| | More Time Spent | −0.11 (0.04) | −2.67 | 664.33 | .008** | [−0.18, −0.03] | | |
| | Training confidence | 0.03 (0.02) | 1.15 | 457.54 | .250 | [−0.02, 0.07] | | |
| | More Time Spent × time$_{TR}$ | 0.03 (0.01) | 2.06 | 327.22 | .040* | [0.00, 0.05] | 0.27 | 0.07 |
| | More Time Spent × time$_{FU}$ | −0.15 (0.07) | −2.22 | 173.46 | .028* | [−0.28, −0.02] | −0.01 | −0.21 |
| RR Positive Bias | Intercept | 2.34 (0.03) | 89.37 | 1,999.70 | <.001*** | [2.29, 2.39] | | |
| | time$_{TR}$ | 0.13 (0.01) | 14.14 | 366.96 | <.001*** | [0.11, 0.15] | | |
| | time$_{FU}$ | −0.21 (0.04) | −5.21 | 217.01 | <.001*** | [−0.29, −0.13] | | |
| | More Time Spent | 0.00 (0.04) | 0.06 | 672.66 | .952 | [−0.07, 0.08] | | |
| | Training confidence | 0.00 (0.02) | 0.07 | 374.71 | .946 | [−0.04, 0.04] | | |
| | More Time Spent × time$_{TR}$ | 0.00 (0.01) | 0.07 | 383.21 | .940 | [−0.03, 0.03] | 0.01 | 0.01 |
| | More Time Spent × time$_{FU}$ | −0.04 (0.07) | −0.67 | 185.97 | .506 | [−0.17, 0.09] | −0.07 | −0.07 |

*Note.* Each outcome was modeled separately. Every model had the fixed effects of engagement group, time$_{TR}$, time$_{FU}$, engagement group × time$_{TR}$, engagement group × time$_{FU}$, training confidence (grand mean centered), a random intercept, and random slopes for time$_{TR}$ and time$_{FU}$ (for random effects, see Table B.12 in S1 File). Engagement group was dummy coded with Less Time Spent as the reference group (0 = Less Time Spent, 1 = More Time Spent). OASIS = Overall Anxiety Severity and Impairment Scale; DASS-21 AS = Depression, Anxiety, Stress Scales-Short Form: Anxiety Subscale; BBSIQ = Brief Body Sensations Interpretations Questionnaire; RR = Recognition Ratings; TR = training trajectory; FU = follow-up trajectory. *p < .05. **p < .01. ***p < .001.
[a]Growth modeling analysis d adjusted for baseline differences. The d next to More Time Spent × time$_{TR}$ is the model-estimated standardized mean difference between groups at Session 5; the d next to More Time Spent × time$_{FU}$ is that at follow-up. These ds are adjusted for any baseline mean differences (and thus reflect differences only in growth, not in initial status). Per Cohen (1988), ds ≥ 0.2, 0.5, or 0.8 are small, medium, or large, respectively [62].
[b]Growth modeling analysis d unadjusted for baseline differences. These ds are unadjusted for any baseline mean differences (and thus reflect differences at Session 5 and at follow-up based on differences in both initial status and growth).

program (including both training and assessment tasks; b = −0.10, p < .001; Table 4) decreased more than the group that spent more time (b = −0.07, p < .001). This slope difference translates to a small standardized mean difference between groups at Session 5 (d adjusted for baseline differences = 0.27; Table 3). The groups' slopes also significantly differed from Session 5 to follow-up (b = −0.15, p = .028; Table 3). The group that spent less time significantly increased in negative bias (b = 0.13, p = .003; Table 4), reflecting some loss in training gains, whereas the group that spent more time had no significant change. This slope difference, combined with the slope difference during the training phase, translates to a negligible mean difference between groups at follow-up (d adj. = −0.01; Table 3).

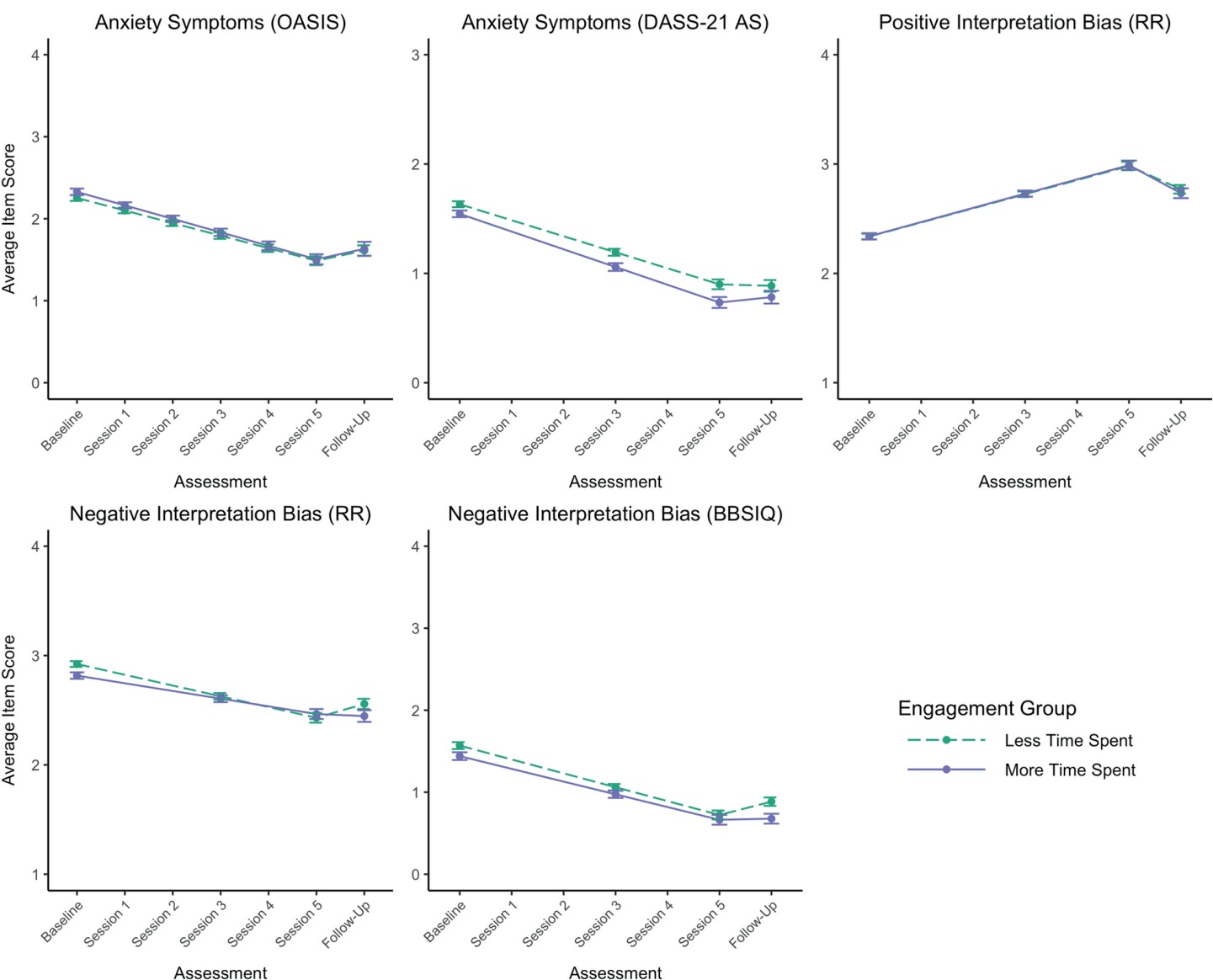

**Fig 2. Model-estimated means over time by engagement group.** *Note.* Estimated means ($\pm 1$ *SE*) from the piecewise linear multilevel models at mean level of training confidence were calculated in each imputed dataset and pooled across datasets using the $\mathrm{testConstraints}$ function of the $\mathrm{mitml}$ package (ver 0.4-3) [66]. Plots were created with the $\mathrm{ggplot2}$ (ver. 3.3.5) [84] and $\mathrm{cowplot}$ (ver. 1.1.1) [85] packages. Estimates are shown only for timepoints at which the measure was assessed. OASIS = Overall Anxiety Severity and Impairment Scale; DASS-21 AS = Anxiety Scale of Depression Anxiety Stress Scales; RR = Recognition Ratings; BBSIQ = Brief Body Sensations Interpretation Questionnaire.

For negative bias assessed by the BBSIQ, unexpectedly, the groups' slopes did not significantly differ during the training phase (Table 3). However, the slopes did significantly differ during the follow-up phase ($b = -0.15$, $p = .033$). As shown in Fig 2, after both groups had reductions in negative bias during the training phase, during the follow-up phase the group that spent less time again significantly increased in negative bias ($b = 0.16$, $p = .001$, Table 4), reflecting some loss in training gains, whereas the group that spent more time had no significant change. The standardized mean difference between the groups at follow-up, based on their similar slopes during the training phase and different slopes during the follow-up phase, was very small (*d* adjusted for baseline differences = -0.09; Table 3).

**Table 4. Piecewise linear multilevel modeling simple time effects for outcomes With significant interaction effects.**

| Outcome | Group | Fixed Effect | b (SE) | t | df | p | 95% CI | d^a |
|---|---|---|---|---|---|---|---|---|
| BBSIQ | Less Time Spent | Time$_{TR}$ | −0.17 (0.01) | −13.32 | 292.37 | <.001*** | [−0.19,−0.14] | −1.01 |
| | | Time$_{FU}$ | 0.16 (0.05) | 3.38 | 168.46 | .001** | [0.07, 0.26] | −0.82 |
| | More Time Spent | Time$_{TR}$ | −0.16 (0.01) | −11.43 | 264.05 | <.001*** | [−0.18, −0.13] | −0.93 |
| | | Time$_{FU}$ | 0.01 (0.05) | 0.27 | 161.03 | .788 | [−0.09, 0.11] | −0.92 |
| RR Negative Bias | Less Time Spent | Time$_{TR}$ | −0.10 (0.01) | −10.05 | 243.67 | <.001*** | [−0.12, −0.08] | −0.92 |
| | | Time$_{FU}$ | 0.13 (0.04) | 3.06 | 175.75 | .003** | [0.05, 0.21] | −0.68 |
| | More Time Spent | Time$_{TR}$ | −0.07 (0.01) | −7.44 | 245.17 | <.001*** | [−0.09, −0.05] | −0.69 |
| | | Time$_{FU}$ | −0.02 (0.05) | −0.38 | 125.70 | .705 | [−0.12, 0.08] | −0.72 |

*Note.* For the simple time effects, separate models were fit for each engagement group with fixed effects for time$_{TR}$, time$_{FU}$, and training confidence (grand mean centered), a random intercept, and random slopes for Time$_{TR}$ and Time$_{FU}$. Simple time effects were analyzed only for significant ($p < .05$) interactions in Table 3. BBSIQ = Brief Body Sensations Interpretations Questionnaire; RR = Recognition Ratings; TR = training trajectory; FU = follow-up trajectory. *$p < .05$. **$p < .01$. ***$p < .001$.
$^a$Growth modeling analysis $d$. The $d$ next to Time$_{TR}$ is the model-estimated standardized mean difference within the group from baseline to Session 5; the $d$ next to time$_{FU}$ is that from baseline to follow-up.

## Discussion

This study analyzed differences in anxiety and interpretation bias outcomes for different behavioral engagement patterns in anxious community adults undergoing web-based CBM-I. *K*-means clustering was used to explore potentially complex engagement patterns based on several program use features involving task completion rate and time spent on training and assessment tasks. Based on this analysis, we created two descriptive engagement groups (likely from a single population vs. two truly distinct subpopulations), with one group generally spending more time on tasks (small-medium effects for training tasks, large effects for assessment tasks). Unexpectedly, participants who generally spent more time on the program (including both training and assessment tasks) did not improve in anxiety and positive interpretation bias significantly more than those who spent less time. Further, although both groups significantly improved in negative interpretation bias during the training phase, participants who spent less time improved in negative bias (when assessed by RR) significantly more than participants who spent more time (and post hoc tests found were significantly older and slightly less educated). However, participants who spent less time had a significant loss in training gains for negative bias (when assessed by RR and BBSIQ) during the follow-up phase, whereas participants who spent more time had no significant change.

### Comparable improvement on anxiety symptoms and positive interpretation bias

Some person-centered DMHI studies have found that groups reflecting higher engagement patterns improve more than other groups [17], and some variable-centered studies of individual metrics of time spent have found that greater time spent is related to better outcomes [26–28,32]. Given these results and the common hypothesis that greater use yields better outcomes [23], after the cluster analysis suggested two groups differing in time spent on tasks, we expected the group that generally spent more time (to the extent that this reflects higher engagement) to improve significantly more than the group that spent less time. However, prior results are mixed, and the comparable improvements we found between the two groups in anxiety and positive interpretation bias in this study are consistent with the results of other person-centered studies in which all engagement groups improved during the intervention, with no group improving significantly more than the others [19,71]. The comparable improvements are also consistent with other variable-centered studies' finding no significant relations between single time spent metrics and outcomes [24,25,30].

### Greater improvement on negative interpretation bias for Less Time Spent

We did find differential improvement between groups for one outcome, but in an unexpected direction: During the training phase, participants who generally spent less time on the program (including both training and assessment tasks) improved in negative interpretation bias (when assessed by RR) significantly more than those who spent more time. Over five sessions, this slope difference translates to a small standardized mean difference between the groups at the end of the training phase. If spending less time is an actual marker of less engagement (e.g., if it indicates reduced intention and interest such that users are skimming vs. reading program materials) [18], then it is intuitive that participants who spend less time may lose some training gains during the follow-up phase (as we found for both measures of negative bias, leading to a negligible mean difference between the groups at follow-up for RR and a very small mean difference between the groups at follow-up for BBSIQ). However, it is unclear why participants who spend less time would improve more in negative bias (or in any outcome) during the training phase than those who spend more time. This finding highlights issues in interpreting time spent as a marker of engagement.

### Issues in measuring engagement

Although spending more time may reflect greater physical energy investment (i.e., behavioral engagement or participation), time spent is a less clear indicator of behavioral investment than the markers used in some prior person-centered studies of engagement groups [17,19,71], such as task completion rate (which was balanced between the groups in our study) and frequency of using the program (which was constrained in our study given that users could not repeat sessions). Effective engagement is investment directed at the focal tasks in the DMHI that are thought to lead to change in outcomes (e.g., targeted mechanisms and downstream symptoms) [22]. For example, in CBM-I, it is primarily the task of vividly imagining emotionally ambiguous scenarios resolving in less threatening ways that is thought to shift interpretation biases and reduce anxiety [8]. However, we do not know exactly how users were spending time, so distraction could have contributed to longer times. Moreover, engagement involves simultaneous *cognitive* (e.g., selective attention to the task) and *affective* (e.g., positive emotional reactions to and commitment to the task) energy investments as well [22]. Thus, although time spent may offer some insight into the process by which users complete tasks, it is critical to understand the actual behaviors and cognitive and affective experiences that underlie users' time spent, while also examining a variety of other markers of behavioral investment (e.g., task completion rates) and its experiential qualities.

### Further characterizing engagement groups

Post hoc tests revealed that the groups significantly differed on aspects other than time spent, providing potential insight into our findings for negative interpretation bias. Notably, in our person-centered approach, any engagement group differences on variables that did not enter into the cluster analysis can help inform interpretation of the groups' engagement patterns. (This contrasts with a variable-centered approach, in which any correlates of a given engagement variable would represent potential confounds of the effect of the engagement variable on outcomes.) Specifically, our post hoc tests raise the possibility that people in the group that generally spent more time on the program, who were significantly older (medium effect) and less educated (very small effect) than people in the group that spent less time, had slower processing speed [72] and lower reading comprehension [73]. These differences would likely add to their time (on both training and assessment tasks, as was observed) and potentially make it more difficult to quickly learn the pattern presented in the training scenarios (that

ambiguous situations tend to end in non-threatening ways), whereas participants who spent less time may have more readily learned this pattern in the short term. However, once users who spent more time had improved, perhaps their gains were maintained at follow-up (unlike users who spent less time, who lost some gains) because of their greater physical investment of time during the training phase. In particular, if users who spent more time thought about the scenarios more deeply or imagined them more vividly, this elaborate processing could have facilitated longer-term recall [39,74,75]. By contrast, if the initial learning for users who spent less time was more superficial, it could have eroded more quickly after the active practice of assigning less negative interpretations during CBM-I had ended.

It is notable that participants who generally spent more time had significantly *lower* anxiety symptoms and negative interpretation bias at baseline (very small to small effects) than participants who spent less time. Moreover, the two groups had comparable depression symptoms, alcohol use, positive interpretation bias, and confidence in the ability of an online training program to reduce their anxiety. The group that spent more time identified with multiple races (vs. one) at a lower rate than the group that spent less time (small effect for race), but the groups did not significantly differ on gender or ethnicity. Although the potential impacts of the group differences in age and education on engagement patterns and treatment outcomes noted above are speculative, they show the value of further characterizing the clusters for clues as to how the clusters might have originated and why the clusters might have some different outcomes.

## Limitations

This study has several limitations. First, given that our homogeneity test did not support rejecting the presence of only a single cluster, our clusters are best interpreted as two descriptive subgroups of one population; we cannot infer the existence of two truly distinct subpopulations [36,57]. Moreover, even though we excluded engagement features that intercorrelated at least .70 from the cluster analysis, the retained features still had non-negligible intercorrelations ($r$s > .1; see Table B.13 in S1 File) that could have contributed to the algorithm's suggesting two clusters with higher versus lower means across most features (i.e., our More Time Spent and Less Time Spent groups, respectively) [36]. Second, our findings about the effects of engagement patterns on outcomes are not causal. Third, our sample was demographically homogeneous, and we may have introduced sampling bias by excluding participants who were assigned to receive supplemental telecoaching. Further, although all participants had at least moderate anxiety symptoms at baseline (and on average had *marked* [76] total OASIS scores [11.55; see Table B.2 in S1 File], surpassing the cutoff of 8 for probable diagnosis [77–79], and *extremely severe* [80] total DASS-21-AS scores [11.27, or 22.54 on the DASS-42-AS]), our results may not generalize to other populations (e.g., those defined by clinical diagnoses; those who would not seek web-based CBM-I). Fourth, the administration of training and assessment tasks in series (e.g., Session 1 assessment was not administered until Session 1 training was finished; vs. in parallel) constrained our selection of engagement markers (e.g., preventing analysis of completion of training tasks separate from assessment tasks), and not all studies include assessment-related markers. Fifth, for markers involving repeated tasks, we used the mean time spent on those tasks across sessions, ignoring likely variability over the study's course; engagement is dynamic and can vary not only between persons, but also within persons [22].

## Future work

Given that engagement is multifaceted, future research on engagement in DMHIs (for both CBM-I and other interventions) should go beyond the typical focus on physical investments

(time spent and other behavioral metrics) [18,20]. Future work could assess affective investments (e.g., perceiving tasks as self-relevant, feeling motivated to do tasks) and cognitive investments (e.g., selectively attending to tasks, using mental resources to meet task demands), in addition to physical investments [22], and study how multifaceted patterns of these investments influence outcomes. Another potentially interesting physical marker is time elapsed between sessions (which we did not include in our cluster analysis because, due to attrition, only 399 participants had the data needed to compute such a variable; see Table B.11 in S1 File). It would also be helpful to directly measure some of the different possible reasons for increased time (e.g., distraction, slower processing speed, doubt about responses). If, as we speculated, users who generally spent less time lost some training gains because they processed the training scenarios less elaborately, future CBM-I programs might include techniques to encourage deeper processing during training (e.g., exercises that apply insights from training to users' own experiences), consolidate gains at end of training (e.g., relapse-prevention exercises), and facilitate skill application after training (e.g., skill reminders, booster sessions). Future work could also identify individual engagement markers that predict outcomes before performing clustering on only the significant predictors [17,30]. To investigate variability in engagement over time, growth mixture models could identify unobserved subgroups from repeated measures of engagement [81,82]. The present study also highlights the value of further characterizing engagement patterns on variables that did not enter into the cluster analysis (e.g., demographic characteristics, symptom severity), a strategy we suggest for future research. Finally, to analyze separate completion rates for training versus assessment tasks, future work should administer these tasks in parallel, allowing users who stop training to continue assessments.

## Conclusion

Understanding differences in treatment outcomes for different real-world patterns of engagement with DMHIs is key to clarifying how and for whom DMHIs are effective and to personalizing DMHIs. This study used an exploratory cluster analysis to create two behavioral engagement groups revealed by program use metrics in a clinical trial of a web-based CBM-I program for anxiety. Unexpectedly, the engagement groups (whose patterns ended up reflecting generally more vs. less time spent across tasks) did not significantly predict differential change in most outcomes. Moreover, participants who generally spent less time on the program (including both training and assessment tasks) improved in one outcome significantly more during the training phase (but had a significant loss in gains for that outcome by 2-month follow-up) than those who spent more time (and post hoc tests found were significantly older and slightly less educated). These unexpected group differences in outcomes highlight the challenge of using time spent metrics as markers of engagement and the need to consider cognitive and affective markers in addition to behavioral markers. Future research on complex engagement patterns in DMHIs holds promise for maximizing their potential to increase access to effective treatment for anxiety.

## Supporting information legends

Supporting information below is available in the online S1 supplement file.

Section A NOTES ON MATERIALS AND METHODS
- A.1 Excluding Telecoaching Participants
- A.2 Cognitive Bias Modification for Interpretation (CBM-I) Details
- A.3 Measure Details and Adaptations

## Acknowledgments

This manuscript is based on AFV's master's thesis [83]. We would like to thank members of the Program for Anxiety, Cognition, and Treatment at the University of Virginia for their comments on an earlier draft of this manuscript; Henry C. Behan for his project coordination; and members of the MindTrails Project research team for their support.

## Author contributions

**Conceptualization:** Ángel Francisco Vela de la Garza Evia, Jeremy William Eberle, Sonia Baee, Bethany Ann Teachman, Laura Elizabeth Barnes.

**Data curation:** Jeremy William Eberle, Sonia Baee.

**Formal analysis:** Ángel Francisco Vela de la Garza Evia.

**Funding acquisition:** Mehdi Boukhechba, Daniel Harold Funk, Bethany Ann Teachman, Laura Elizabeth Barnes.

**Methodology:** Ángel Francisco Vela de la Garza Evia, Jeremy William Eberle, Sonia Baee.

**Project administration:** Ángel Francisco Vela de la Garza Evia.

**Resources:** Jeremy William Eberle.

**Supervision:** Laura Elizabeth Barnes.

**Visualization:** Ángel Francisco Vela de la Garza Evia.

**Writing – original draft:** Ángel Francisco Vela de la Garza Evia, Jeremy William Eberle, Emma Catherine Wolfe.

**Writing – review & editing:** Ángel Francisco Vela de la Garza Evia, Jeremy William Eberle, Sonia Baee, Emma Catherine Wolfe, Mehdi Boukhechba, Bethany Ann Teachman, Laura Elizabeth Barnes.

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
