## [Decision Letter · Decision Letter 0]

16 Mar 2025

PDIG-D-25-00096Behavioral engagement patterns and psychosocial outcomes in web-based interpretation bias training for anxietyPLOS Digital Health Dear Dr. Teachman, Thank you for submitting your manuscript to PLOS Digital Health. After careful consideration, we feel that it has merit but does not fully meet PLOS Digital Health's publication criteria as it currently stands. Therefore, we invite you to submit a revised version of the manuscript that addresses the points raised during the review process. Please submit your revised manuscript within 60 days May 15 2025 11:59PM. If you will need more time than this to complete your revisions, please reply to this message or contact the journal office at digitalhealth@plos.org. Please include the following items when submitting your revised manuscript:* A rebuttal letter that responds to each point raised by the editor and reviewer(s). You should upload this letter as a separate file labeled 'Response to Reviewers'. This file does not need to include responses to any formatting updates and technical items listed in the 'Journal Requirements' section below.* A marked-up copy of your manuscript that highlights changes made to the original version. You should upload this as a separate file labeled 'Revised Manuscript with Track Changes'.* An unmarked version of your revised paper without tracked changes. You should upload this as a separate file labeled 'Manuscript'. If you would like to make changes to your financial disclosure, competing interests statement, or data availability statement, please make these updates within the submission form at the time of resubmission. Guidelines for resubmitting your figure files are available below the reviewer comments at the end of this letter. We look forward to receiving your revised manuscript. Kind regards, Haleh AyatollahiSection EditorPLOS Digital Health Leo Anthony CeliEditor-in-ChiefPLOS Digital Healthorcid.org/0000-0001-6712-6626 **Journal Requirements:** **Additional Editor Comments (if provided):****Reviewers' Comments:** Reviewer's Responses to Questions

**Comments to the Author**

1. Does this manuscript meet PLOS Digital Health’s publication criteria? Is the manuscript technically sound, and do the data support the conclusions? The manuscript must describe methodologically and ethically rigorous research with conclusions that are appropriately drawn based on the data presented.

Reviewer #1: Yes

Reviewer #2: Yes

Reviewer #3: Yes

2. Has the statistical analysis been performed appropriately and rigorously?

Reviewer #1: Yes

Reviewer #2: Yes

Reviewer #3: Yes

3. Have the authors made all data underlying the findings in their manuscript fully available (please refer to the Data Availability Statement at the start of the manuscript PDF file)?

Reviewer #1: Yes

Reviewer #2: Yes

Reviewer #3: Yes

4. Is the manuscript presented in an intelligible fashion and written in standard English?

Reviewer #1: Yes

Reviewer #2: Yes

Reviewer #3: Yes

5. Review Comments to the Author

Reviewer #1: 1. The revisions made in response to the previous reviewers' feedback are commendable. These improvements have significantly enhanced the clarity and quality of the work, making its findings more effectively communicated.

2. The unexpected finding that lower-engagement users initially improved more in negative interpretation bias but later lost these gains is interesting, Do you have anything to provide a compelling theoretical explanation for this pattern

3.Overall, this is a well-executed study with meaningful implications for enhancing online mental health treatments. It makes a significant contribution to the evolving field of digital mental health interventions.

4. The research is well-structured, methodologically robust, and provides valuable insights for clinicians, researchers, and developers of digital mental health programs.

5. The study does not sufficiently address potential confounding factors that could explain why participants who spent less time showed initial benefits but lost them at follow-up?

6. This abstract is well-structured and clearly outlines the key objectives regarding behavioral engagement patterns and psychosocial outcomes in web-based interpretation bias training for anxiety.

7. Future interventions should incorporate cognitive markers (e.g., attention, cognitive load) and affective markers (e.g., motivation, task relevance). Future studies should also examine how these factors influence engagement and long-term outcomes.

Reviewer #2: I appreciate the valuable contribution this study makes to understanding engagement patterns in digital mental health interventions, particularly in the context of web-based cognitive bias modification for interpretation (CBM-I) in anxious adults. One of the most important insights from this research is that measuring engagement purely in terms of time spent may be insufficient for optimizing DMHI effectiveness, which has significant implications for intervention design and evaluation.

The authors have also done an excellent job in addressing the reviewers' previous comments, providing thorough explanations and clarifications that have strengthened the manuscript. However, I recommend that they acknowledge the limitations of their sample, as it consists of self-selected anxious adults in an online setting. This may affect the generalizability of the findings, particularly for clinical populations or individuals with lower digital literacy. Additionally, while the study convincingly argues that time spent is an inadequate engagement marker, it would be helpful to suggest concrete strategies for improving CBM-I interventions. For instance, should engagement be adaptive, with task length adjusting based on user progress? Addressing these aspects would further enhance the manuscript's clarity and practical relevance.

Reviewer #3: Thank you for the opportunity to review this manuscript. It presents an interesting analysis and take on engagement with a web-based training program for anxiety. I have very few comments, I think it is very well written and clearly explained and really only have a couple of thoughts that I felt were missing from the discussion.

1. I understand your rationale for performing your analysis this way and using the clusters to explore complex patterns of engagement. After reading the manuscript, I wondered whether this is the best way to explore engagement? Given no differences and the difficulties interpreting time spent, on reflection would task completion actually have given a better sense of how much people actually worked through the content? Since you don't seem to have done any post-hoc analysis looking at this, or refer to other work that did do this with this data, I felt this could be discussed in the discussion.

2. Similarly although you talk about the differences between groups which is very interesting, you don't do any post-hoc analysis controlling for these. Again I felt maybe this was missing in the discussion, whether these are worth further exploration in analysis, how they might be impacting outcomes.

6. PLOS authors have the option to publish the peer review history of their article (what does this mean?). If published, this will include your full peer review and any attached files.

**Do you want your identity to be public for this peer review?** For information about this choice, including consent withdrawal, please see our Privacy Policy.

Reviewer #1: **Yes: **Bala Nimmana

Reviewer #2: No

Reviewer #3: No

---

## [Decision Letter · Decision Letter 1]

27 Jun 2025

Behavioral engagement patterns and psychosocial outcomes in web-based interpretation bias training for anxiety

PDIG-D-25-00096R1

Dear Prof Teachman,

We are pleased to inform you that your manuscript 'Behavioral engagement patterns and psychosocial outcomes in web-based interpretation bias training for anxiety' has been provisionally accepted for publication in PLOS Digital Health.

Best regards,

Haleh Ayatollahi

Section Editor

PLOS Digital Health

**Additional Editor Comments (if provided):**

**Reviewer Comments (if any, and for reference):**

Reviewer's Responses to Questions

**Comments to the Author**

1. If the authors have adequately addressed your comments raised in a previous round of review and you feel that this manuscript is now acceptable for publication, you may indicate that here to bypass the “Comments to the Author” section, enter your conflict of interest statement in the “Confidential to Editor” section, and submit your "Accept" recommendation.

Reviewer #1: All comments have been addressed

Reviewer #2: All comments have been addressed

Reviewer #3: All comments have been addressed

2. Does this manuscript meet PLOS Digital Health’s publication criteria? Is the manuscript technically sound, and do the data support the conclusions? The manuscript must describe methodologically and ethically rigorous research with conclusions that are appropriately drawn based on the data presented.

Reviewer #1: Yes

Reviewer #2: Yes

Reviewer #3: Yes

3. Has the statistical analysis been performed appropriately and rigorously?

Reviewer #1: Yes

Reviewer #2: Yes

Reviewer #3: Yes

4. Have the authors made all data underlying the findings in their manuscript fully available (please refer to the Data Availability Statement at the start of the manuscript PDF file)?

Reviewer #1: Yes

Reviewer #2: Yes

Reviewer #3: Yes

5. Is the manuscript presented in an intelligible fashion and written in standard English?

Reviewer #1: Yes

Reviewer #2: Yes

Reviewer #3: Yes

6. Review Comments to the Author

Reviewer #1: 1. The authors have addressed all prior reviewer comments with detailed and thoughtful responses, demonstrating clear engagement with the feedback.

2. The revised Discussion appropriately clarifies why demographic variables were not statistically controlled, aligning with a person-centered analytic framework.

3. The authors have added concrete suggestions for future CBM-I program adaptations

Reviewer #2: The suggested revisions have made the manuscript substantially more robust, and I recommend it for publication.

Reviewer #3: I have no further comments to add, thank you for addressing them all in prior rounds thoroughly.

7. PLOS authors have the option to publish the peer review history of their article (what does this mean?). If published, this will include your full peer review and any attached files.

**Do you want your identity to be public for this peer review?** For information about this choice, including consent withdrawal, please see our Privacy Policy.

Reviewer #1: **Yes: **Bala Nimmana

Reviewer #2: None

Reviewer #3: **Yes: **Lauren Jerome
